# The Transcriptional Landscape of Immune-Response 3′-UTR Alternative Polyadenylation in Melanoma

**DOI:** 10.3390/ijms25053041

**Published:** 2024-03-06

**Authors:** Xiao Yang, Yingyi Wu, Xingyu Chen, Jiayue Qiu, Chen Huang

**Affiliations:** Dr. Nesher’s Biophysics Laboratory for Innovative Drug Discovery, State Key Laboratory of Quality Research in Chinese Medicine, Macau University of Science and Technology, Taipa, Macao SAR 999078, China; 2109853cct20001@student.must.edu.mo (X.Y.); viviennewu77@outlook.com (Y.W.); xiaozeshiwoxiaodi@outlook.com (X.C.); 3230006748@student.must.edu.mo (J.Q.)

**Keywords:** alternative polyadenylation, melanoma, tumor microenvironment, immunotherapy

## Abstract

The prognosis of patients with malignant melanoma has been improved in recent decades due to advancements in immunotherapy. However, a considerable proportion of patients are refractory to treatment, particularly at advanced stages. This underscores the necessity of developing a new strategy to improve it. Alternative polyadenylation (APA), as a marker of crucial posttranscriptional regulation, has emerged as a major new type of epigenetic marker involved in tumorigenesis. However, the potential roles of APA in shaping the tumor microenvironment (TME) are largely unexplored. Herein, we collected two cohorts comprising melanoma patients who received immune checkpoint inhibitor (ICI) immunotherapy to quantify transcriptome-wide discrepancies in APA. We observed a global change in 3′-UTRs between responders and non-responders, which might involve DNA damage response, angiogenesis, PI3K-AKT signaling pathways, etc. Ten putative master APA regulatory factors for those APA events were detected via a network analysis. Notably, we established an immune response-related APA scoring system (IRAPAss), which exhibited a great performance of predicting immunotherapy response in multiple cohorts. Furthermore, we examined the correlation of APA with TME at the single-cell level using four single-cell immune profiles of tumor-infiltrating lymphocytes (TILs), which revealed an overall discrepancy in 3′-UTR length across diverse T cell populations, probably contributing to immunoregulation in melanoma. In conclusion, our study provides a transcriptional landscape of APA implicated in immunoregulation, which might lay the foundation for developing a new strategy for improving immunotherapy response for melanoma patients by targeting APA.

## 1. Introduction

Melanoma is one of the most common cancers, with rising incidence and mortality. The majority of cases tend to occur cutaneously, referred to as skin melanoma (SKCM), which has been regarded as the most aggressive and fatal subtype of skin tumor [1]. Melanoma pathogenesis has been demonstrated to be associated with genetic alterations. However, growing evidence has disclosed a complex involvement of epigenetic mechanisms, including methylation, chromatin modification, and remodeling [2]. Aberrant DNA methylation and conformational changes in chromatin via the post-translational modification of histones have been relatively well studied in melanoma tumorigenesis [3,4]. Alternative polyadenylation (APA), a unique posttranscriptional regulatory mechanism to determine the length of the 3′-untranslated region (3′-UTR), has widely been found to be associated with the tumorigenesis of diverse cancers [5,6,7]. However, the potential roles of APA in tumorigenesis, particularly crosstalk with the tumor microenvironment (TME) in melanoma, has not been comprehensively investigated yet.

In addition, immune checkpoint inhibitor (ICI) immunotherapy is a very promising treatment strategy extensively used in a variety of cancer therapies, particularly for melanoma. However, approximately 40–50% of melanoma patients have no response to ICI immunotherapy [8]. Recent studies have revealed that APA is intricately involved in the regulation of antitumor immunotherapy responses and immunity [9,10]. Typically, programmed death ligand 1 (PD-L1), one of the most common targets in ICI immunotherapy, was reported to assist escaping CTL-mediated immune surveillance by shortening the length of its 3′-UTR [11]. Therefore, it is rational to hypothesize that targeting on APA or relevant regulatory pathways might affect the TME, by which ICI immunotherapy response in melanoma patients may be substantially improved. Meanwhile, on the basis of immune-related APA events, the establishment of a model for predicting immunotherapy response in melanoma is practicable. Herein, we performed a systematic transcriptomic analysis of the aberrance in 3′-UTR length based on RNA-seq data, by which we identified trends in APA impacting the immunity of melanoma, and unveiled for the first time differential APA events probably implicated in immunotherapy response in melanoma, as well as corresponding master APA regulatory factors. The similar results and APA heterogeneity across different T cells were further revealed at a single-cell resolution. Our study shed light on the underlying mechanisms of APA in tumors against host immunity, which may provide new insights into improving immunotherapy strategies for melanoma patients.

## 2. Results

### 2.1. Landscape of Differential APA Events between Responders and Non-Responders in Melanoma

To investigate potential APA events correlated to ICI immunotherapy in melanoma, we used a well-established algorithm DaPars2 to identify the dynamic APA events based on two melanoma immunotherapy cohorts (GSE78220 and GSE91061). The difference of PDUI score profiles between responders and non-responders in respective cohorts was assessed according to a previous study [12] (for details, see Section 4). This analysis disclosed distinct differential APA events exhibited in two cohorts (Figure 1A,B, Appendix A), suggesting a substantial heterogeneity of APA events in melanoma patients. We then performed unsupervised clustering using a hierarchical clustering method based on the PDUI profile of differential APA events, respectively (Figure 1C). The results showed that the responders (PRCR) and non-responders (PDSD) could be well separated in both cohorts, which suggested that the PDUI score might be a useful predictor of the immunotherapy response in melanoma patients.

Next, we seek to examine the potential functions of those differential APA events. The genes found to have APA events were subjected to Metascape for enrichment analysis (Figure 1D,E, Appendix A). The 3′-UTR shortening genes were found to be involved in many pathways related to tumorigenesis and immunity, i.e., DNA damage response, angiogenesis, PI3K-AKT signaling pathways, the regulation of interferon beta production, etc. The function of genes with lengthened 3′-UTR refers to vascular development and cell division. In line with the presence of common APA events across multiple cancer types identified in previous studies [5,13], the alterations in the 3′-UTR might link to cancer metabolism, e.g., glycine, serine, and threonine metabolism (Figure 1D).

### 2.2. Master APA Factors Mediating APA Events and Probable Impact on Immunity in Melanoma Patients

The 3′ end-processing machinery refers to multiple APA factors, including CPSF, CSTF, CFI, etc. [14,15,16]. To explore which APA factor(s) probably mediate those differential APA events between responders and non-responders in melanoma, we recruited 98 key APA factors via literature screening (Appendix A) and performed a correlation analysis between those APA factors and differential APA events. In addition, the protein–protein interactions (PPIs) between the genes were found to have differential APA events, and these APA factors were extracted from STRING database, aiming to obtain more potential APA regulatory patterns. Then, combined with the interactions previously achieved by correlation analysis, a differential APA events to APA factors (ATF) network was initially established, which comprised 94 APA factors and 63 differential APA events (Figure 2A).

Additionally, the network topology analysis was conducted via Cytoscape, which achieved 10 master APA factors, including U2AF2, SF3B1, DDX39B, etc. (Figure 2B). Notably, the majority of differential APA events were co-regulated by multiple master APA factors (Figure 2C,D and Appendix A). Typically, DHX8, an ATP-dependent RNA helicase that regulates the release of spliced mRNAs from spliceosomes prior to their export from the nucleus, was found in the present study to be jointly regulated by five master APA factors and exhibited a shortened 3′-UTR length in non-responders. Recent studies have disclosed the potential roles of helicases in innate immunity, i.e., DHX15, DHX16, DHX29, etc. [17]. Certainly, the associations between DHX8 and immunomodulation in melanoma requires further validation.

We next investigated the impact of the master APA factors on the TME via four algorithms, respectively, including ImmuneScore, ESTIMATEscore, TIDEscore, and IPScore. The master APA factors in non-responders exhibited significantly high correlation with those immune scores compared with that in responders, suggesting that those master APA factors might play roles in negatively regulating immunity in melanoma (Figure 2E). To validate this view, we further examined the correlation between those master APA factors and immune checkpoint genes, which represent a class of immunosuppressive molecules and have been demonstrated to be closely associated to ICI immunotherapy response to some degree. The results showed that the majority of master APA factors exhibited extraordinary correlation with immune checkpoints genes in non-responders compared with that in responders (Figure 2F and Appendix A), i.e., PD-L1, CD47, and PDCD1LG2. These results are consistent with our previous correlation analysis between master APA factors and immune scores. In addition, we used the random forest to examine the impact of these APA factors on immune checkpoint genes, which revealed that DDX39B was the main contributor to the expression of CD80 and CD209. DHX15 might involve PD-L1, CD47, and PDCD1LG2 (Figure 2F and Appendix A).

### 2.3. Establishment of the IRAPAss Model for Predicting the Response to Immunotherapy in Melanoma Patients

Increasing evidence has shown that APA events can be used as prognostic markers for cancer [5,18,19,20,21]. However, few reports have explored whether they can be used as potential biomarkers to predict the response of tumor patients to immunotherapy [22]. Herein, we proposed an in silico analysis pipeline. Using the GSEA analysis enrichment results, we identified key APA events associated with immune-related pathways with *pathAPAscore* > 0.995 and FDR < 0.05. These events showed opposing directions of enrichment between responders and non-responders (Figure 3A; for the detailed procedure, see Section 4). A total of 1518 APA event–pathway pairs and 271 APA events were identified. Next, based on these 271 APA events, we used a univariate Cox regression and LASSO Cox regression analysis to screen out 10 immunotherapy response-related APA events associated with survival (including AIM2|chr1|159062567, BAX|chr19|48960961, COL27A1|chr9|114310700, etc.) (Figure 3B), which were further subjected to a neural network model to establish an immunotherapy response-related APA scoring system (IRAPAss). IRAPAss exhibited a decent predictive capability of immunotherapy response (Figure 3C) and prognosis (Figure 3D). Concretely, the responders exhibited a higher IRAPAss score than that in non-responders (Figure 3E–H and Appendix A).

We further verified the robustness and effectiveness of the IRAPAss using independent cohorts. As a result, IRAPAss exhibited a fairly good performance in predicting immunotherapy response and prognosis (Figure 3I and Appendix A). In line with what we observed in the training and testing cohorts (randomly split metaAPA data (70% versus 30%)), the patients with a higher IRAPAss score exhibited a better prognosis and more likely response to ICI immunotherapy (Appendix A). In conclusion, IRAPAss can serve as a good marker for predicting immunotherapy response in melanoma.

### 2.4. Correlation Analysis between IRAPAss and TME

Next, we investigated the underlying mechanism by which IRAPAss predicts immunotherapy. The correlation of IRAPAss with many immune scores was initially assessed in multiple cohorts. We observed that the patients with a higher IRAPAss score exhibited a higher proportion of B cells, memory B cells, and effector memory CD8+ T cells (Figure 4A), whereas a lower level of MHC, IFN-γ, TLS, and T cell-inflamed scores were observed (Figure 4B) compared with that in patients with lower IRAPAss scores.

Meanwhile, a multivariate Cox regression analysis was performed based on several factors related to OS, including immunotherapy response, IRAPAss (Group), StromalScore, and ImmuneScore. The results indicated that IRAPAss was an independent prognostic predictor (Figure 4C and Appendix A). Then, we created nomograms to predict OS. The results revealed that non-responders had a total score of 1.29 and a 91.2% OS rate after 633 days (Figure 4D), illustrating an exceptional predictive ability of IRAPAss. Similar results were found in the other two cohorts (Appendix A).

Next, we probed the potential correlation of key APA events of IRAPAss with the TME via a page ranking network algorithm (for details, see Section 4). The top 200 genes with high impact scores were retained (Appendix A). A subsequent functional analysis suggested that they were associated with RNA processing and transport, protein processing and modification, and the cell cycle, as well as many pathways related to immunotherapy response (Figure 4E), e.g., the activation of immunotherapy response, T cell proliferation, the regulation of activated T cell proliferation, the regulation of T cell cytokine production, etc. Taken together, these analyses suggested that IRAPAss could reflect the heterogeneity and differences of the TME for melanoma patients.

### 2.5. Master APA Factors Are Prone to Diverging across Distinct T Cell Populations in Melanoma

To unveil the heterogeneity of APA events across different cell types impacting the TME in melanoma, we integrated single-cell immune profiles of TILs from four cohorts for further analysis (Appendix A). After quality control, 186,227 cells were categorized into 34 clusters using UMAP (Appendix A). Then, those clusters were annotated into eight populations according to the expression of markers of well-known T cell types (Figure 5A,B, Appendix A), including naïve T cells (Tns), antigen-presenting CD8+ T cells (APCs), proliferactive CD8+ T cells, effector memory T cells (Tems), cytotoxic CD8+ T cells (Tcs), T helper 17 cells (Th17s), terminally differentiated effector memory cells (TEMRAs), exhausted CD8+ T cells (Texs), and tumor regulatory CD4+ T cells (Tregs). These T cells exhibited distinct proportions across responders and non-responders. In addition, we classified the T cell types into two subgroups (immunosuppression and immunoactivation; for details, see Section 4). The non-responders had high proportions of immunosuppression populations (multiple chi-square test, *p* < 0.001; Figure 5C). A similar phenomenon was observed in our previous analysis in bulk RNA-seq data (Figure 5D,E). Furthermore, we compared the expression of ten master APA factors between two subgroups. The results showed that most of them were differentially expressed between the two populations (Figure 5F), yet no significant difference was observed in the bulk RNA-seq data (Figure 5G and Appendix A).

### 2.6. Global Discrepancy in 3′-UTR Length across Diverse T Cell Populations May Contribute to Immunoregulation in Melanoma

To investigate whether the heterogeneity of APA events impacts the TME of melanoma patients at a single-cell resolution, we initially detected APA events based on scRNA-seq data. Meanwhile, we categorized the patients into three subgroups according to the proportion of subpopulations of T cells, including T cell activation (TA), T cell depletion (TS), and T cell naive (TN) (Figure 6A, Appendix A). As expected, the majority of responders were clustered into TA, and non-responders were predominantly aggregated into TS. Then, a differential APA analysis was conducted among three subgroups (TN vs. TA + TS, TA vs. TN + TS, TS vs. TN + TA), which disclosed a widespread difference of 3′-UTR among those subgroups (Figure 6B,C and Appendix A). Additionally, the potential impact of these altered APA events was partially validated at the protein level via the TCGA-SKCM cohort (Appendix A).

In addition, we identified several dynamic APA events with gradient PDUI scores across three subgroups (Figure 6D). For example, overexpression of myc-associated zinc finger protein (MAZ) was reported to drive peroxisome proliferator-activated receptor γ1 (PPARγ1) expression, which involves tumorigenesis and progression [23,24]. Subsequently, the genes found to have those APA events were subjected to Metascape for functional enrichment analysis (Figure 6E, Appendix A). Several pathways, including the apoptotic signaling pathway and immune system development, were enriched in TA. TS referred to phosphorylation, the regulation of cell activation and growth, PD-L1 expression, and the PD-1 checkpoint pathway, whereas TN exhibited a higher activity in cell cycle, adaptive immune response, and the regulation of the MAPK cascade, etc. Through a comparison between TA and TS, independently, we identified 120 representing 3′-UTR shortening and 108 representing 3′-TUR lengthening APA events (Figure 6F, Appendix A). The functional enrichment analysis showed that the APA events of 3′-UTR shortening were specifically involved in response to tumor necrosis factor and natural killer cell-mediated cytotoxicity (Figure 6G, Appendix A).

Next, we examined the potential roles of APA factors during the developmental trajectories of T lymphocyte subpopulations; a trajectory analysis via Slingshot was initially performed, which yielded four trajectories with a common starting point of Tns (Figure 6H). Specifically, the putative trajectories begin by starting with Tns and pass through Th17s and Tems, ending with four distinct cell types, including TMERAs (trajectory 1), proliferactive CD8+ T cells (trajectory 2), Tregs (trajectory 3), and Texs (trajectory 4). Notably, many differentially expressed APA factors between the T cell activation trajectory (trajectory 2) and the T cell exhaustion trajectory (trajectories 3 and 4) were identified (Appendix A). We further explored which APA events might be regulated by those APA factors by a network analysis (Figure 6I, details see Section 4). As a result, those 31 APA factors involved 8272 APA events. In particular, those APA events might participate in diverse immune-related pathways, i.e., adaptive immune response, Th17 cell differentiation, inflammatory response, antigen processing, and presentation (Figure 6I, Appendix A). More importantly, many master APA factors, which has been figured out by our previous study, were also identified at the single-cell level, including U2AF2, DHX8, and TRIM5 (Figure 6J and Appendix A). In conclusion, our analysis firstly disclosed a global difference of APA events mediated by APA factors, which probably contribute to immunity regulation in melanoma.

## 3. Discussion

ICI immunotherapy is a promising cancer treatment strategy, but nearly half of melanoma patients do not benefit from immunotherapy. Recent advances in transcriptome analysis have showed that many genes related to immune response are affected by RNA posttranscriptional regulatory mechanisms [25,26]. APA, a pre-RNA processing mechanism widespread in all eukaryotes, is one of the major mechanisms of gene regulation [27]. APA has been reported to exhibit a quite good predictive power of prognosis [5], but thus far, regarding immunotherapy, only a small number of APA events were identified as immunotherapy responses of modulation [22].

This study examined the capacity of APA events to anticipate immune responses in melanoma patients undergoing immunotherapy. A total of 10 APA events were identified as immunotherapy-related biomarkers in the present study. Among the parental genes for these APA events, some are related to tumorigenesis or immunity. For example, AIM2 is a member of the interferon (IFN)-induced PYHIN protein family, whose main function is to sense pathogen-associated or host-derived cytosolic dsDNA, recruit other inflammatory components, and induce caspase-dependent inflammation. In addition, AIM2 is also important for the normal function of regulatory T cells. BAX itself is a proapoptotic protein that can participate in the regulation of apoptosis. On the basis of those APA events, we successfully constructed a model, namely IRAPAss, to predict immunotherapy response, which exhibited a good performance in predicting immunotherapy response and prognosis in both the training cohort and independent validation datasets.

Recently, oncologists have been focusing on the discovery of effective biomarkers for predicting outcomes and guiding precision medicine [28,29,30]. APA has been reported to have a strong predictive ability for the prognosis of cancer patients [6,16,18]. However, relatively few APA events have been reported for the prediction of immunotherapy [5], especially in melanoma. Our study disclosed several APA events that are probably implicated in immunotherapy response in melanoma. For example, for PPP3B, identified by our differential APA analysis, when its transcription was repressed, it was found to inhibit the increase in p-FOXO3a levels after transcription suppression, thereby destroying the negative feedback inhibition loop formed by FOXO3a and miRNA; this led to the upregulated expression of IGF2 and IRS1. In turn, IGF2 and IRS1 are significantly associated with poor prognosis in breast cancer patients [31]. OTULIN is an inverse regulator of the NF-κB pathway, which is associated with inflammatory problems. OTULIN-deficient cells, whose Met1 UB chain overaccumulates on the target substrates IKK/NEMO, RIRK, and ASC, will overproduce many crude inflammatory cytokines, compromising the therapeutic effect of cytokine inhibitors [32].

Additionally, we examined the possible implications of APA in TME at the single-cell level. We found that non-responders exhibited a higher proportion of Th17 cells and a lower proportion of APC and TEM cells than responders in melanoma, which was in line with our previous bulk RNA-seq analysis. The dominant perception is that Th17 cells and Th17-related cytokines are positively correlated to tumor immunity, but few studies have reported that they may have tumor-promoting abilities [33,34]. The high cell ratio of Th17 compared with the low cell ratio of APC may be one of the reasons for melanoma patients’ resistance to immunotherapy. The patients were divided into three subgroups (TA, TS, and TN) based on distinct proportions of T cells. Among them, many differential APA events were identified. For instance, specific APA events in TA may involve apoptotic signaling pathways, the regulation of macroautophagy, and immune system development. For TS, the APA events were found to be related to the regulation of cell growth and PD-L1 expression and the PD-1 checkpoint pathway in cancer. These findings are consistent with our results at the bulk RNA-seq level, indicating that aberrant APA events may contribute to immune dysfunction by the regulation of gene expression and cellular processes. In particular, many altered APA events were further validated at the protein level. For instance, the protein expression of INPP4B was observed to be lowly expressed in the non-responders (TCGA-SKCM). Notably, INPP4B was found to serve as a tumor suppressor gene in melanoma via regulating PI3K/Akt signaling, which impacts the proliferative, invasive, and tumorigenic capacity of melanoma cells [35]. Our APA analysis based on scRNA-seq data disclosed that INPP4B exhibited a shortened 3′-UTR in non-responders. Generally, the shortening of the 3′-UTR was associated with gene overexpression due to its avoidance of miRNA-mediated repression. This paradox might be explained by the fact that gene expression regulation is not only limited to APA, but also involves complex processes, i.e., DNA methylation, transcription factor, RNA editing, microRNAs and long noncoding RNAs, etc. Certainly, further experiments might be required to validate this finding.

Apart from those key APA events, their master APA factors were identified through the network analysis. These APA factors play important roles in RNA cleavage, processing, and translocation. We found that the expression of master factors mostly differed significantly between IA and IS at the single-cell resolution. Certainly, further validations were required to unveil the mechanism underlying how APA factors regulate APA events.

Our study has many shortcomings. First, all findings of this study were discovered based on a computational methodology; a further ‘wet lab’ approach should be applied to validate the key conclusions in the future. Second, this study included a limited number of patients that received immunotherapy; it is essential to recruit more samples for analysis.

In summary, our study firstly characterized the differences in the APA aspects of responders and non-responders in melanoma patients that received immunotherapy via a systemic transcriptomic perspective and revealed the impact of APAs mediated by APA factors on the TME in melanoma patients. These findings provide new insights for alleviating the immunosuppressive microenvironment of non-responders and provide a viable strategy for improving the efficacy of immunotherapy by targeting APA.

## 4. Materials and Methods

### 4.1. Data Resources

Four cohorts of melanoma patients who received ICI immunotherapy from the ENA database (https://www.ebi.ac.uk/ena/browser/view/) were used in the present study, including GSE78220 (13 PDSD and 14 PRCR), GSE91061 (82 PDSD and 23 PRCR), PRJEB23709 (42 PDSD and 49 PRCR), and GSE135222 (19 PDSD and 8 PRCR). The gene expression profile of 102 cutaneous melanoma cases (TCGA-SKCM) and protein profiles of 38 cutaneous melanoma cases were obtained from the TCGA database (https://portal.gdc.cancer.gov/) (Appendix A). We collected the single-cell immune profiling of tumor-infiltrating lymphocytes (TILs) of 30 melanoma patients from four cohorts (GSE232444, GSE211504, GSE242477, and GSE229858). Of the patients, 13 received immunotherapy, including 6 PRCR and 7 PDSD (Appendix A).

### 4.2. Data Pre-Processing

The raw data of bulk RNA-seq was filtered using Trimmomatic v.0.39 [36] to eliminate low-quality reads and adapters and further aligned to the human genome (GRCh38) of Ensemble database via STAR v.2.7.9a [37]. The gene expression profile quantified via featurecounts [38] was transferred into TPM using an in-house R script. The bedGraph and wig files generated using BEDTools v.2.30.0 [39] and Samtools v.1.16.1 [40], respectively, were used as the input of DaPars2 v.2.1 [6,41] for APA detection. The 3′-UTR differences of transcripts were quantified as a PDUI score via DaPars2 v.2.1 [5,18].

The Cell Ranger v7.2.0 pipeline was used to process the raw data of single-cell RNA-seq with default parameters. Gene-barcode matrices were generated for each sample by counting the unique molecular identifiers (UMIs) and barcode count, and genes without expression across all cells were removed. Cells with expressed genes of <200 or >6000 were excluded. The percentage of UMI mapped to mitochondria and hemoglobin was set below 25% and 0.1%.

### 4.3. Differential APA Events Detection and Downstream Analysis

The PDUI scores of all samples within the same group were averaged and named as Mean_PDUI_GroupA/B/C. The difference in mean 3′-UTR change per transcript between two groups was quantified by ΔPDUI (ΔPDUI = Mean_PDUI_GroupA − Mean_PDUI_GroupB). A permutation test with looser conditions was used to control the false discovery rate (FDR < 0.05). The differential APA events were defined according to two criterions: ① |ΔPDUI| ≥ 0.1; ② *p* value < 0.05.

The cluster analysis of the PDUI profiles was conducted using the ComplexHeatmap package [42,43], and the functional analysis of differential APA events was achieved by using Metascape v3.5 [44].

A total of 98 APA factors screened out via literature retrieval [10,45] were subjected to a Spearman correlation analysis with differential APA events, which aimed at detecting possible APAfactors. Additional interactions among them were extracted from the STRING database v.11.5 [46], and the network was visualized using Gephi v.0.10.1 [47]. A network topology analysis was conducted using 12 algorithms (including Betweenness, BottleNeck, Closeness, ClusteringCoefficient, Degree, DMNC, EcCentricity, EPC, MCC, MNC, Radiality, and Stress) via cytoHubba [48] in Cytoscape v.3.9.1 [49].

### 4.4. Computational Assessment of Immune Microenvironment

Infiltrating immune cell fractions were estimated via CIBERSORT, and Xcell, and several immune-related scores were used to assess the immunity of the tumor microenvironment, respectively, including the ImmuneScore, ESTIMATEscore, TIDEscore, IPScore, ESTIMATE, TIDE, IPS, and IOBR packages [50,51,52,53,54].

### 4.5. Establishment of an Immunotherapy Response-Related APA Scoring System (IRAPAss)

We built a scoring system to predict immunotherapy response for patients according to Gaoyang Wang’s study [22]. Concretely, we initially screened out potential APA events related to immune-related pathways based on three gene sets (hallmark.all. v.2022.1, c2.cp.kegg. v.2022.1, c2.cp.reactome. v.2022.1) from the Molecular Signatures Database (MSigDB) [55,56,57]. A GSEA analysis was conducted to assess the activity of pathways via the clusterProfiler package [58]. Then, we linked APA events and pathways via a Pearson correlation analysis. Highly correlated APA events were retained to establish the model, where the detailed procedure was conducted as follows.

First, all samples could be divided into two groups (responders and non-responders) according to their immunotherapy clinical information. The expression of gene *x* and APA events *y* across all samples were defined as G(*x*) = (g_1_, g_2_, g_3_, …, g_n_), and A(*y*) = (a_1_, a_2_, a_3_, …, a_n_), respectively. Next, for each pair of gene and APA events, we calculated the correlation (Cor) between the gene expression (G) and PDUI score of the APA event (A) using a Pearson correlation analysis, which yielded a *p* value of Cor, defined as P(*xy*). For each pair of gene and APA event, a correlation score (*CS*) was defined as follows:$$\;CS\left(xy\right)=-log10(P(xy)\;{+10}^{-20})\;*\;\;sign(Cor\left(xy\right))$$

Then, on the basis of three gene sets of MSigDB, we used the ranked *CS* value as the input for GSEA analysis to figure out immune-related APA events. The enrichment score (*ES* (*y*, *h*)) and *p* value (*p*) between APA event *y* and related pathway h were subjected to a calculation of the *pathAPAscore* as follows:$$\;pathAPAscore\left(y,h\right)=\left\{\;\begin{array}{c} 1-2p,\;\mathrm{if}\;ES\left(y,h\right)>0\;\\ 2p-1,\;ES\left(y,h\right)<0 \end{array}\;\right.$$

We screened out significant APA pathway pairs (*pathAPAscore* > 0.995 and FDR < 0.05). Then, we compared APA pathway pairs with responders and non-responders based on the normalized enrichment score (NES) derived from previous GSEA analysis. The APA pathway pairs exhibiting the opposite NES value between responders and non-responders were screened out. These series of analyses were conducted in two melanoma cohorts (GSE78220 and GSE91061), and the APA pathway pairs detected in both cohorts with the same orientation were retained for further modeling.

Afterwards, in the GSE91061 cohort, we filtered out the specific APA events significantly associated with tumor prognosis via a univariate Cox hazard analysis and LASSO Cox regression (*p* < 0.05) based on the survival data. Finally, we combined two melanoma cohorts (GSE78220 and GSE91061), which were referred to as metaAPA, randomly split metaAPA data into the training and testing data (70% versus 30%), and established a model using a multilayer perceptron (MLP) [59] via the TensorFlow package, as follows:$$IRAPAss=0.785057366\;*\;AIM2\left|chr1\right|159062567\;+0.660980225\;*\;BAX\left|chr19\right|48960961\;-0.564149737\;*\;COL27A1\left|chr9\right|114310700\;-0.060074512\;*\;CTTN\left|chr11\right|70422660\;-0.037628233\;*\;ERH\left|chr14\right|69380489\;-1.267010808\;*\;GTF3C2-AS1\left|chr2\right|27337588\;+0.819018066\;*\;NASP\left|chr1\right|45618212\;+0.974817872\;*\;RPL13\left|chr16\right|89563035\;-0.624692976\;*\;TAF15\left|chr17\right|35847057\;-0.279070169\;*\;TMEM63A|chr1|225841036$$

### 4.6. Braided Target–Proximal Gene PPI Network

We initially extracted the human PPI network from the STRING database v.11.5 (interaction scores greater than 700; contains 16,584 nodes and 252,833 edges) [60,61]. Then, we put all the features (the genes found to have APA events) that were used to construct the IRAPAss into the PPI network to calculate the maximum connectivity component using NetworkX and identify the feature-based modules via the page rank algorithm [62] with the default setting (damping factor = 0.85). The top 200 genes with the highest impact scores were regarded as the feature-based module.

### 4.7. Single-Cell RNA-Seq Analysis

Four cohorts were combined using the Harmony package to eliminate the batch effects. The ScaleData function was used to identify variable features using default parameters. A Principal component analysis (PCA) was performed on the scaled data to reduce the dimensionality. The first 20 principal components were used for identifying the neighbors and clustering the cells with a resolution of 0.9. The cell clusters were visualized using 2D uniform manifold approximation projection (UMAP) plots. We used the FindAllmarkers feature of the Seurat package to identify markers (log2 fold change > 0.25 and adjusted *p* value < 0.05). The Slingshot package was used for pseudo-temporal analysis and visualization [63].

### 4.8. Statistical Analysis

All statistical analyses were performed via R v4.2.2 (two-sided, *p* value < 0.05 considered statistically significant). A univariate Cox regression analysis and multivariate Cox regression analysis were used to screen the features. A survival analysis was conducted using the Survival package. A nomogram was established to assess whether IRAPAss was an independent factor [64]. The performance of the model was assessed via the AUC value using the pROC package.

## Figures and Tables

**Figure 1 ijms-25-03041-f001:**
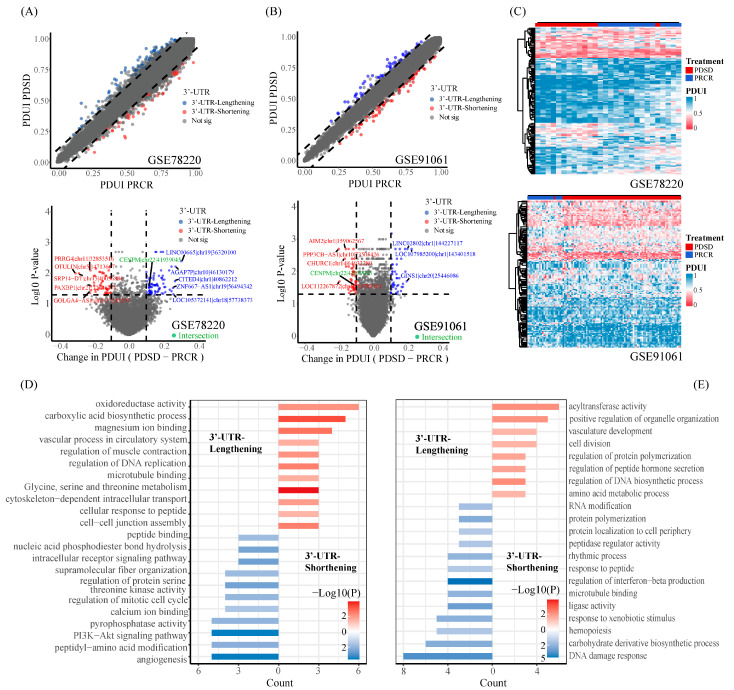
The comprehensive analysis of RNA-seq data identified 3′-UTR alterations associated with immunotherapy response. (**A**,**B**) The average PDUI score curve and volcano plot for each APA event in the PRCR group and the PDSD group in the GSE78220 cohort and GSE91061 cohort. The dashed line indicates the cutoff at 0.1. Blue dots represent genes with lengthened 3′-UTRs, and red dots represent genes with shortened 3′-UTRs. The green color represents the intersection of APA events in the two patient cohorts. (**C**) Heatmap showing genes (row) undergoing 3′-UTR shortening (red) or 3′-UTR lengthening (blue) in each melanoma patient (column) receiving immunotherapy. (**D**) Metascape enrichment results of differential APA events in the GSE78220 cohort. (**E**) Metascape enrichment results of differential APA events in the GSE91061 cohort.

**Figure 2 ijms-25-03041-f002:**
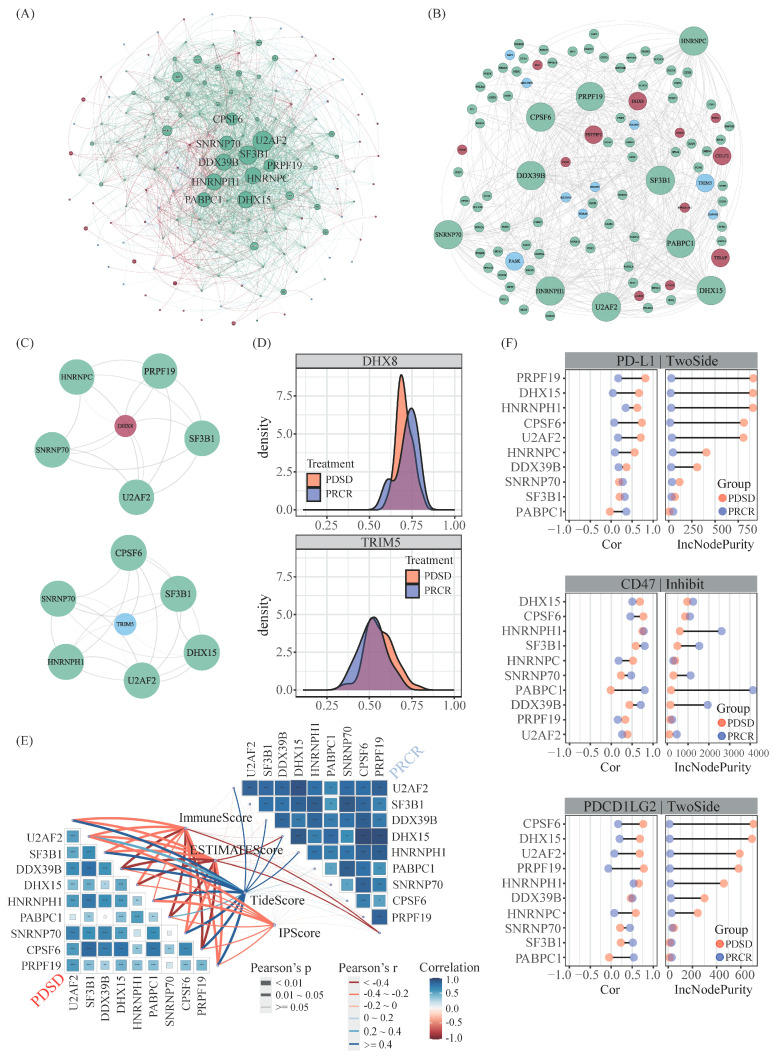
Regulatory network and further analysis. (**A**) The regulatory network constituted by APA factors and the parental genes for differential APA events. The blue node represents genes with 3′-UTR lengthening, the red node represents genes with 3′-UTR shortening, and the green node represents APA factors. The red lines indicate positive correlations between the PDUI score of parental genes for differential APA events and the expression of APA regulatory factors, while the blue lines indicate negative correlations. The green line represents the protein–protein interactions both between the parental genes for differential APA events and APA factors and within themselves. (**B**) Subnetwork of masterAPA factors. (**C**) The DHX8 gene and TRIM5 gene are all co-regulated by multiple master APA factors. (**D**) The PDUI scores of APA events occurring in the DHX8 gene, TRIM5 gene in different groups. (**E**) Butterfly plot of correlations between master APA factors and four immune-related scores. The blue indicates a positive correlation, while the red indicates a negative correlation. The *p* values of the figures are shown as follows: *, *p* < 0.05. **, *p* < 0.01. ***, *p* < 0.001. (**F**) Correlation analysis of CD209 gene expression with 10 master APA factors and the degree of contribution of these 10 master APA factors.

**Figure 3 ijms-25-03041-f003:**
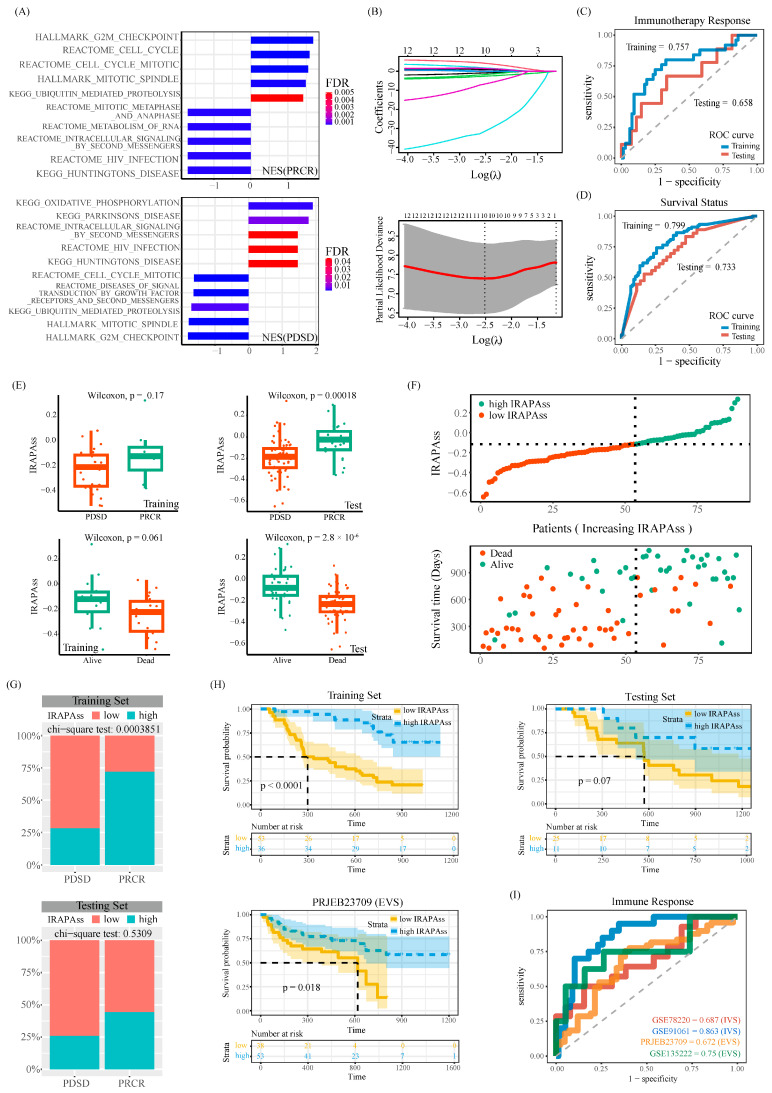
Establishment of the IRAPAss. (**A**) The enrichment score distribution of APA events in the two cohorts, taking the biological pathways with the opposite direction; each line represents a biological pathway. (**B**) LASSO Cox analysis results of OS-related APA events. Each line of different colors represents an APA event. (**C**) ROC curve of the immunotherapy response signature. (**D**) ROC curve of the OS signature. (**E**) Boxplots of IRAPAss distributions for responders and non-responders and surviving and deceased patients in the training and testing data. (**F**) Dot plot. According to the optimal cutoff value, the samples were divided into two groups: one with a higher IRAPAss score and one with a lower IRAPAss score. (**G**) Histogram of the distribution of samples with a higher IRAPAss score and samples with a lower IRAPAss score between responders and non-responders in the training data and testing data. (**H**) The Kaplan–Meier plot of IRAPAss. (**I**) ROC curves of immunotherapy response signatures in four patient cohorts.

**Figure 4 ijms-25-03041-f004:**
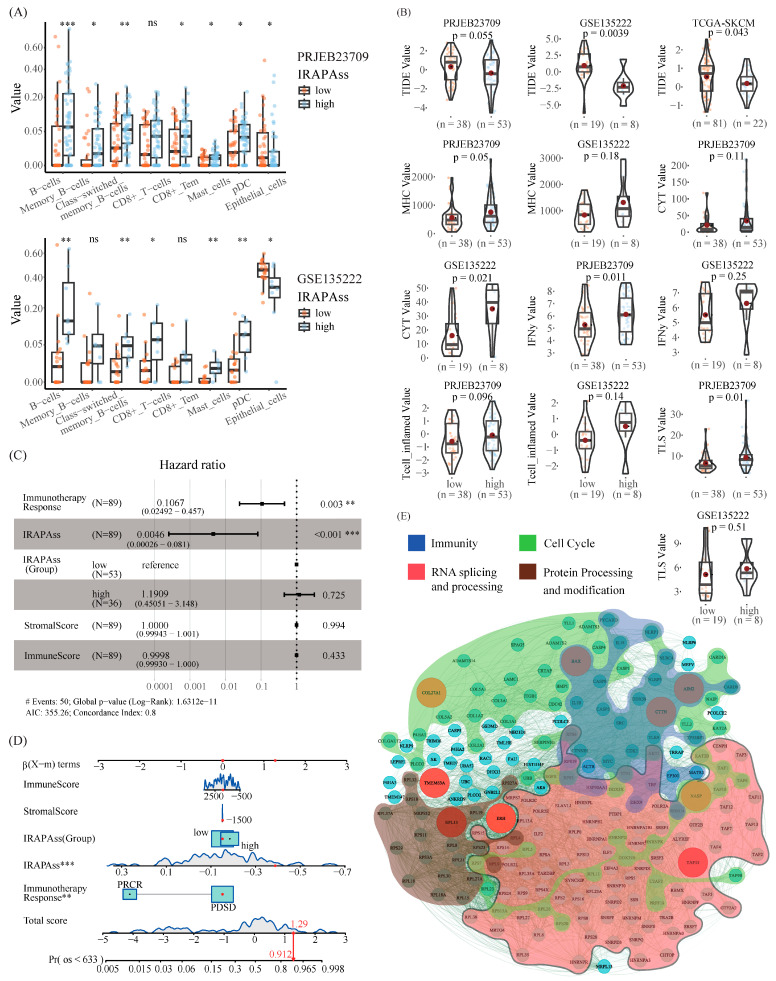
Immunization-related evaluation based on IRAPAss. (**A**) Differences in immune cell and stromal cell scores between samples with a higher IRAPAss score and samples with a lower IRAPAss score. (**B**) Differences in immune-related scores between samples with a higher IRAPAss score and samples with a lower IRAPAss score. (**C**) Multivariate Cox regression analysis of IRAPAss and melanoma clinical data. (**D**) Nomograms containing IRAPAss and prognostic clinical data were built to predict the OS of melanoma patients. (**E**) Construction of a PPI network based on the proximal genes of the model signatures. The *p* values of the figures are shown as follows: ns, *p* > 0.05. *, *p* < 0.05. **, *p* < 0.01. ***, *p* < 0.001.

**Figure 5 ijms-25-03041-f005:**
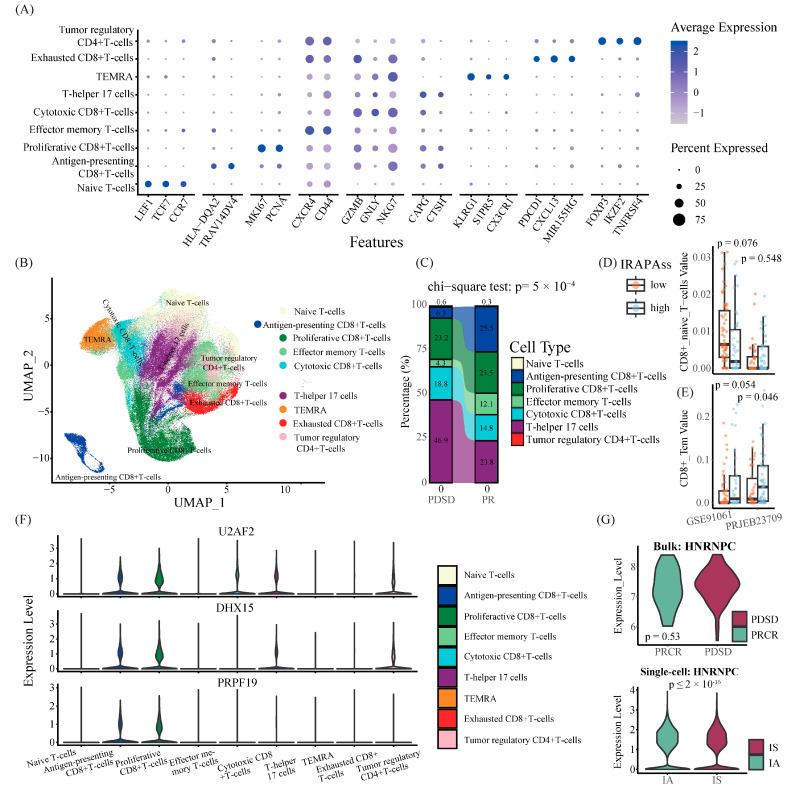
Single-cell transcriptome profiles of melanoma patients. (**A**) Expression pattern of marker genes for each cell cluster. (**B**) The uniform manifold approximation projection (UMAP) plots of the cell types and resources profiled in this study. (**C**) Sankey diagram showing the fraction of each cell type between PDSD and PRCR groups. (**D**) Boxplot demonstrating the difference in CD8+_Naive_T cell values between samples with a higher IRAPAss score and samples with a lower IRAPAss score. (**E**) Boxplot demonstrating the difference in CD8+_Tem values between samples with a higher IRAPAss score and samples with a lower IRAPAss score. (**F**) Violin plots demonstrating the expression levels of APA factors across different T cell types. (**G**) Violin plots demonstrating the expression levels of APA factors in the PDSD/PRCR group, as well as in the IA/IS cell population. *p* values were calculated by a two-sided Wilcoxon rank sum test.

**Figure 6 ijms-25-03041-f006:**
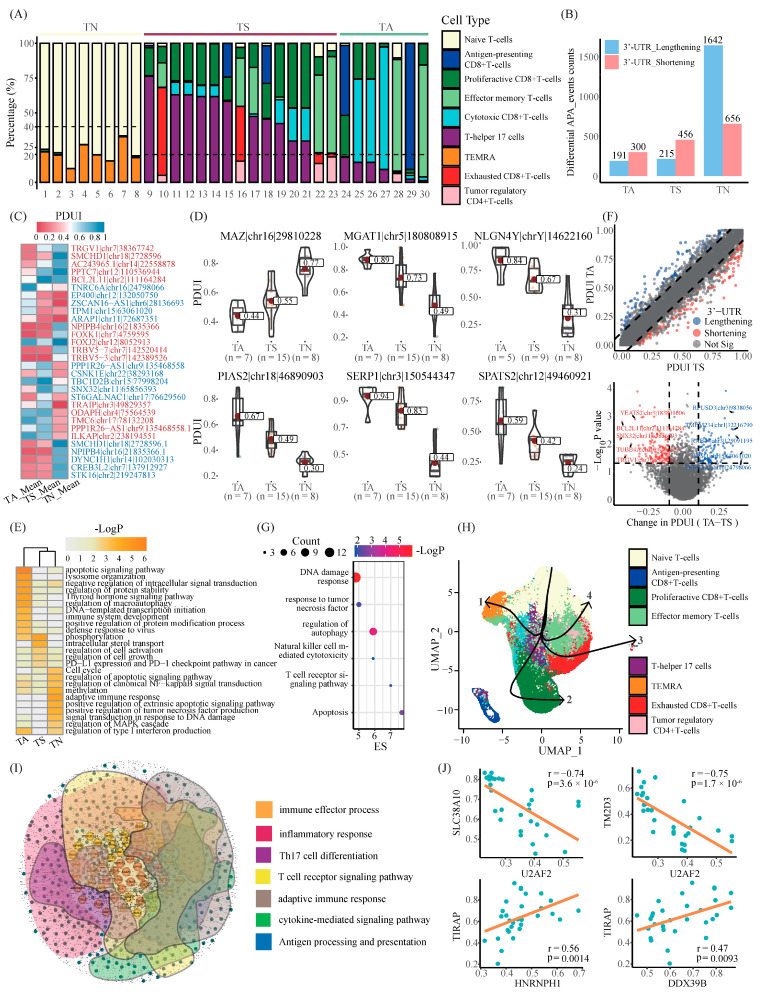
Comprehensive analysis of differential APA events in TN, TS, and TA populations. (**A**) The bar plot displays the proportion of cells in each T cell subpopulation in TN, TS, and TA clusters from 30 samples. TN clusters are represented by light yellow, TS clusters by reddish brown, and TA clusters by a grey-green color. (**B**) The bar plot displays the number of differential APA events in the TN, TS, and TA clusters. 3′-UTR shortening APA events are shown in light red, and 3′-UTR lengthening APA events are shown in light blue. (**C**) The heatmap displays the average PDUI scores for each of the five APA events, comparing the smallest and largest Group_Diff_PDUI values in the TN, TS, and TA clusters. (**D**) The violin plots illustrate the six APA events in which PDUI scores either increased or decreased along the gradient in the TA, TS, and TN clusters. (**E**) The heatmap displays the outcomes of the enrichment analysis of differential APA events with shortened 3′-UTR in the TA, TS, and TN clusters. (**F**) The results of the analysis of variance between TA clusters and TS clusters are presented in the form of mean PDUI score curves and volcano plots for each APA event. The 0.1 critical point is indicated by the dashed line. Genes with 3′-UTR lengthening are represented by blue dots, while those with 3′-UTR lengthening are represented by red dots. (**G**) The bubble chart displays the outcomes of the enrichment analysis of differential APA events with shortened 3′-UTR in TA–TS. (**H**) The differentiation trajectory of T cells on the steady state map was represented using a master curve, with the root being the naive T cell (Tn) clusters. (**I**) The network diagram illustrates the correlation analysis between candidate APA factors and parental genes for APA events. The red nodes represent master APA factors, the yellow nodes represent parental genes for core APA events, and the green nodes represent parental genes for immune-related APA events. (**J**) The dot plots illustrate the correlation trends between master APA factors and APA event parental genes that are identical to the subnetwork.

## Data Availability

All processed data can be accessed from the article and Appendix A or by contacting the corresponding author. All real datasets used within this study are retrievable from public databases with the details provided within the Appendix A.

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
