# Peer review of "The Transcriptional Landscape of Immune-Response 3′-UTR Alternative Polyadenylation in Melanoma"

_ijms, 2024, doi:10.3390/ijms25053041_

Round 1
Reviewer 1 Report
Comments and Suggestions for Authors
I have reviewed the article "The transcriptional landscape of immune-response 3′UTR alternative polyadenylation in melanoma" presented by Yang et al. The article effectively highlights the improved prognosis of malignant melanoma patients due to advancements in immunotherapy over the past decade. It appropriately addresses APA as a crucial posttranscriptional regulatory mechanism involved in tumorigenesis, highlighting its emergence as an important epigenetic marker. This lays the groundwork for investigating its potential role in shaping the tumor microenvironment (TME) and influencing treatment response. The study's methodology involving the analysis of two cohorts of melanoma patients receiving immune checkpoint inhibitor (ICI) immunotherapy provides a robust framework for quantifying transcriptome-wide discrepancies in APA between responders and non-responders. This approach allows for the identification of global changes in 3’-UTRs associated with treatment response, and the detection of ten putative master APA regulators through network analysis adds depth to the study by providing insights into the molecular mechanisms underlying APA-mediated immunoregulation in melanoma. Overall, the study's findings provide a comprehensive transcriptional landscape of APA implicated in immunoregulation, laying the foundation for developing novel therapeutic strategies aimed at improving immunotherapy response in melanoma patients by targeting APA. This highlights the translational potential of the research findings and their significance for future clinical interventions.
Here are some suggestions for additional analyses that could strengthen and expand the understanding of the findings:
What in vitro and in vivo experiments could be considered to validate the functional effects of the identified APA events in regulating immune response and sensitivity to immunotherapy treatment in melanoma models?
It would be important to complement transcriptomic analyses with protein expression studies to verify if alterations in APA events translate into corresponding changes in the production of proteins associated with identified biological processes and signaling pathways.
Investigate the exact function of identified APA regulators through gene silencing or overexpression assays to determine how they specifically influence APA events regulation and immune response in the context of melanoma.
Explore interactions between host cells (such as tumor-infiltrating immune cells) and tumor cells about APA patterns using co-expression and spatial co-localization analysis approaches in tumor tissue samples. Assess the ability of APA profiles and the IRAPAss scoring system to predict immunotherapy treatment response in other cancer types, which could provide insights into the generalization of these findings beyond melanoma.
The authors should explore if changes in APA events are associated with the development of acquired resistance to immunotherapy treatment in initially sensitive patients, through the analysis of tissue samples before and after disease progression.
These additional analyses would further contextualize the study's findings and advance understanding of the mechanisms regulating immune response and treatment resistance in melanoma.
Author Response
Reviewer#1:
Reply: Thank you for your kind comments. Your comments are very important and valuable to improve the quality of this paper. We have carefully read your questions and suggestions and have explained each one and revised the manuscript respectively.
Question 1: What in vitro and in vivo experiments could be considered to validate the functional effects of the identified APA events in regulating immune response and sensitivity to immunotherapy treatment in melanoma models?
Reply: Thank you very much for your valuable and helpful comments. To determine the true impact of the identified APA events on the 3'UTR changes of their corresponding transcripts, 3'RACA assays can be employed. Additionally, a cell model can be created using a melanoma cell line, where the parental genes of the known APA events will be silenced or knocked out, and the growth of the cell will be observed. Finally, the cells are transplanted into mice to construct a melanoma model, and the model is used in conjunction with immunotherapy to observe the growth of the tumor. We hope to implement these analyses with the help of some partners or platforms in our future work, to validate and extend our findings more deeply. We would like to thank you again for your patience and professionalism, and we hope that you will understand our dilemma and give us some tolerance and support.
Question 2: It would be important to complement transcriptomic analyses with protein expression studies to verify if alterations in APA events translate into corresponding changes in the production of proteins associated with identified biological processes and signaling pathways.
Reply: Thank you very much for your valuable and helpful comments. We agree with you and the potential impact of altered APA events has been partially validated by protein expression studies through protein expression data from the TCGA-SKCM cohort. Here's an additional description of the content from the resubmitted article:
Original text: Page 10, row 250
“
Then a differential APA analysis was conducted among three subgroups (TN vs. TA + TS, TA vs. TN + TS, TS vs. TN + TA), which disclosed a widespread difference of 3’-UTR among those subgroups (Figure 6B, C, Supplementary Figure S12 and Supplementary Table S13-15).
”
Revised: Page10, row 258
“
Then a differential APA analysis was conducted among three subgroups (TN vs. TA + TS, TA vs. TN + TS, TS vs. TN + TA), which disclosed a widespread difference of 3’-UTR among those subgroups (Figure 6B, C, Figure S12 and Table S14-16). Additionally, the potential impact of these altered APA events was partially validated at protein level via TCGA-SKCM cohort (Figure S13).
”
Original text: Page 13, row 358
“
These findings are consistent with our results at the bulk RNA-seq level, indicating that aberrant APA events may contribute to immune dysfunction by the regulation of gene ex-pression and cellular processes.
”
Revised: Page13, row 367
“
These findings are consistent with our results at the bulk RNA-seq level, indicating that aberrant APA events may contribute to immune dysfunction by the regulation of gene ex-pression and cellular processes. In particular, many altered APA events were further vali-dated at protein level. For instance, the protein expression of INPP4B was observed to be lowly expressed in the non-responders (TCGA-SKCM). Notably, INPP4B was found to serve as a tumor suppressor gene in melanoma via regulating PI3K/Akt signaling, which impacts the proliferative, invasive, and tumorigenic capacity of melanoma cells [1]. Our APA analysis based on scRNA-seq data disclosed that INPP4B exhibited a shortened 3'-UTR in non-responders. Generally, the shortening of the 3'-UTR is associated with gene overexpression due to its avoidance of miRNA-mediated repression. This paradox might be explained by the fact that gene expression regulation is not only limited to APA, but al-so involves complex processes, i.e., DNA methylation, transcription factor, RNA editing, microRNAs and long noncoding RNAs, etc. Further experiments, certainly, might be re-quired to validate this finding.
”
Original text: Page 14, row 382
“
Gene expression profile of 102 Cutaneous Melanoma (TCGA-SKCM) was obtained from the TCGA database (Supplementary Table S1).
”
Revised: Page14, row 404
“
Gene expression profile of 102 Cutaneous Melanoma (TCGA-SKCM) and protein profiles of 38 Cutaneous Melanoma were obtained from the TCGA database (Table S1-2).
”
Question 3: Investigate the exact function of identified APA regulators through gene silencing or overexpression assays to determine how they specifically influence APA events regulation and immune response in the context of melanoma.
Reply: Thank you very much for your valuable and helpful comments. We agree with you that it would be very meaningful and valuable to investigate the exact function of identified APA regulators through gene silencing or overexpression assays. Due to the limitations of our research conditions, we are currently unable to conduct these analyses. In our subsequent research, we aim to conduct these analyses in collaboration with partners or platforms to further corroborate and enrich our conclusions.
Question 4: Explore interactions between host cells (such as tumor-infiltrating immune cells) and tumor cells about APA patterns using co-expression and spatial co-localization analysis approaches in tumor tissue samples. Assess the ability of APA profiles and the IRAPAss scoring system to predict immunotherapy treatment response in other cancer types, which could provide insights into the generalization of these findings beyond melanoma.
Reply: Thank you for your feedback and suggestions. We appreciate your valuable input regarding the need for additional experiments. However, our current laboratory resources, including equipment, materials, and personnel, are fully allocated to ongoing projects, making it challenging to accommodate additional experiments within the given timeline. We understand the importance of robust scientific research and acknowledge the potential value of the suggested experiments. If future opportunities arise, we will consider incorporating them into our investigations.
The results section description could be improved. The GSE13522 cohort consisted of 27 patients with non-small cell lung cancer who received immunotherapy, as shown in Table S1. This demonstrates the effectiveness of the IRAPAss scoring system in predicting response to immunotherapy treatment in other cancer types.
Original text: Page 7, row 178
“
In conclusion, IRAPAss can serve as a good marker for predicting immunotherapy re-sponse in melanoma.
”
Revised: Page7, row 184
“
In line with what we observed in the training and testing cohorts (randomly split metaAPA data (70% versus 30%)), the patients with a higher IRAPAss score exhibit a better prognosis and more likely response to ICI immunotherapy (Figure S7). In conclusion, IRAPAss can serve as a good marker for predicting immunotherapy response in melanoma.
”
Question 5: The authors should explore if changes in APA events are associated with the development of acquired resistance to immunotherapy treatment in initially sensitive patients, through the analysis of tissue samples before and after disease progression.
Reply: Thank you very much for your valuable and helpful comments. We tried to collect and examine multiple cohorts from GEO (GSE79668, GSE79671, GSE98394, GSE115821, GSE126044), but we found, in all cohorts, that the clinical information (i.e., immunotherapy outcomes) of tissue samples from the same patients before and after immunotherapy treatment were identical. Thus, we cannot perform this analysis based on current data. We expect to collect additional cohorts to further extend our findings in future work.
- Perez-Lorenzo, R.; Gill, K. Z.; Shen, C.-H.; Zhao, F. X.; Zheng, B.; Schulze, H.-J.; Silvers, D. N.; Brunner, G.; Horst, B. A., A tumor suppressor function for the lipid phosphatase INPP4B in melanocytic neoplasms. Journal of Investigative Dermatology 2014, 134, (5), 1359-1368.
Reviewer 2 Report
Comments and Suggestions for Authors
In this study, Yang et al performed the computational analyses of alternative polyadenylation regulation in melanoma patient samples. They found that APA profiles of some genes show different patterns between responders vs. non-responders to immunotherapy. They established a scoring system named IRAPAss which can predict immunotherapy response. They further used single-cell RNA-seq data to show the correlation between IRAPAss vs. tumor immune-composition. The work is novel as no prior study has been performed analyzing APA in melanoma immunotherapy. And the findings are interesting. I just have a few minor comments.
1. For Figure 1C, the values shown in the heatmap appear to be the absolute PDUI values. It is better to show the relative values (log2) across patient samples so that readers can understand which ones show lengthening or shortening in responders vs. non-responders.
2. For the results section, the authors tend to write very short with only the conclusions but did not describe the analysis steps and detailed results in each figure panel. It is hard for readers to understand how they obtain the conclusions. Although the authors describe them in the figure legends and methods sections, they can add more details to the results section. For example, in Figure 3, what are genes used to build IRAPAss, what are the rationales, and how do they obtain the list of genes? The authors introduced analysis steps in the methods section. These should be briefly described in the results section. Another example is that they did not describe the training and testing groups in Figure 3. This causes the inconvenience for readers to understand what analyses they performed. The authors can consider expanding their results section with more detailed descriptions for figure panels.
Author Response
Reviewer#2:
Reply: Thank you for your kind comments. Your comments are very important and valuable to improve the quality of this paper. We have carefully read your questions and suggestions and have explained each one and revised the manuscript respectively.
Question 1: For Figure 1C, the values shown in the heatmap appear to be the absolute PDUI values. It is better to show the relative values (log2) across patient samples so that readers can understand which ones show lengthening or shortening in responders vs. non-responders.
Reply: Thank you very much for your valuable and helpful comments. Gene expression values are typically normalised when heatmapping is performed to present results clearly. As mentioned in the literature cited in section 4.2 of the article, the PDUI scores used to measure the length of the 3'-UTR have been normalised to a range of 0 to 1. A score closer to 0 indicates a shortened 3'-UTR, while a score closer to 1 indicates a lengthened 3'-UTR. Therefore, the PDUI scores in the heatmap do not need further normalisation (log2).
[1]
Question 2: For the results section, the authors tend to write very short with only the conclusions but did not describe the analysis steps and detailed results in each figure panel. It is hard for readers to understand how they obtain the conclusions. Although the authors describe them in the figure legends and methods sections, they can add more details to the results section. For example, in Figure 3, what are genes used to build IRAPAss, what are the rationales, and how do they obtain the list of genes? The authors introduced analysis steps in the methods section. These should be briefly described in the results section. Another example is that they did not describe the training and testing groups in Figure 3. This causes the inconvenience for readers to understand what analyses they performed. The authors can consider expanding their results section with more detailed descriptions for figure panels.
Reply: Thank you very much for your valuable and helpful comments. We provide additional details in the results and methods sections to enhance the reader's understanding of the content.
Original text: Page 5, row 152
“
Herein, we proposed an in-silico analysis pipeline to screen key APA events correlated to immune-related pathways, which exhibited opposite enrichment direction between re-sponders and non-responders (Figure 3A, detailed procedure see Methods). A total of 1518 APA event-pathway pairs and 271 APA events were identified. Next, we used univariate cox regression and the LASSO-Cox regression analysis to screen out 10 immune re-sponse-related APA events associated with survival (Figure 3B), which was further sub-jected to neural network model to establish an immune response-related APA scoring sys-tem (IRAPAss).
”
Revised: Page 5, row 153
“
Herein, we proposed an in-silico analysis pipeline. Using the GSEA analysis enrichment results, we identified key APA events associated with immune-related pathways with pathAPAscore > 0.995 and FDR < 0.05. These events showed opposite directions of enrichment between responders and non-responders (Figure 3A, detailed procedure see Methods). A total of 1518 APA event-pathway pairs and 271 APA events were identified. Next, based on these 271 APA events, we used univariate cox regression and the LASSO-Cox regression analysis to screen out 10 immunotherapy response-related APA events associated with survival (including AIM2|chr1|159062567, BAX|chr19|48960961, COL27A1|chr9| 114310700, etc) (Figure 3B), which was further subjected to neural network model to establish an immunotherapy response-related APA scoring system (IRAPAss).
”
Original text: Page 7, row 176
“
In line with what we observed in the training and test cohorts, the patients with a higher IRAPAss score exhibit a better prognosis and more likely response to ICI immunotherapy (Supplementary Figure S6-7).
”
Revised: Page7, row 184
“
In line with what we observed in the training and testing cohorts (randomly split metaAPA data (70% versus 30%)), the patients with a higher IRAPAss score exhibit a better prognosis and more likely response to ICI immunotherapy (Figure S7).
”
Original text: Page14, row 448
“
These series of analyses were conducted in two melanoma cohorts and the APA-pathway pairs detected in both cohorts with the same orientation were retained for further modeling.
Afterwards, we filtered out the specific APA events significantly associated with tumor prognosis via univariate cox hazard analysis and Lasso-Cox regression (P < 0.05) based on the survival data. Finally, multilayer perceptron (MLP) was used to establish model [60] via TensorFlow package, as follows:
”
Revised: Page15, row 473
“
These series of analyses were conducted in two melanoma cohorts (GSE78220 and GSE91061) and the APA-pathway pairs detected in both cohorts with the same orientation were retained for further modeling.
Afterwards, in the GSE91061 cohort, we filtered out the specific APA events significantly associated with tumor prognosis via univariate cox hazard analysis and Lasso-Cox regression (P < 0.05) based on the survival data. Finally, we combined two melanoma cohorts (GSE78220 and GSE91061) which were referred to as metaAPA, randomly split metaAPA data into the training and testing data (70% versus 30%), and established a model using a multilayer perceptron (MLP) [61] via TensorFlow package, as follows:
”
- Xia, Z.; Donehower, L. A.; Cooper, T. A.; Neilson, J. R.; Wheeler, D. A.; Wagner, E. J.; Li, W., Dynamic analyses of alternative polyadenylation from RNA-seq reveal a 3'-UTR landscape across seven tumour types. Nat Commun 2014, 5, 5274.